# Hypoxia induction in cultured pancreatic islets enhances endothelial cell morphology and survival while maintaining beta-cell function

**Krishana S. Sankar[1,2], Svetlana M. Altamentova[2], Jonathan V. Rocheleau**[1,2,3,4]*

**1** Department of Physiology, University of Toronto, Toronto, Ontario, Canada, **2** Toronto General Hospital Research Institute, University Health Network, Toronto, Ontario, Canada, **3** Institute of Biomaterials and Biomedical Engineering, University of Toronto, Toronto, Ontario, Canada, **4** Department of Medicine, University of Toronto, Toronto, Ontario, Canada

* jon.rocheleau@utoronto.ca

## Abstract

### Background

Pancreatic islets are heavily vascularized *in vivo* yet lose this vasculature after only a few days in culture. Determining how to maintain islet vascularity in culture could lead to better outcomes in transplanting this tissue for the treatment of type 1 diabetes as well as provide insight into the complex communication between beta-cells and endothelial cells (ECs). We previously showed that islet ECs die in part due to limited diffusion of serum albumin into the tissue. We now aim to determine the impact of hypoxia on islet vascularization.

### Methods

We induced hypoxia in cultured mouse islets using the hypoxia mimetic cobalt chloride (100 µM $CoCl_2$). We measured the impact on islet metabolism (two-photon NAD(P)H and Rh123 imaging) and function (insulin secretion and survival). We also measured the impact on hypoxia related transcripts (*HIF-1α*, VEGF-A, *PDK-1*, *LDHA*, C*OX4*) and confirmed increased VEGF-A expression and secretion. Finally, we measured the vascularization of islets in static and flowing culture using PECAM-1 immunofluorescence.

### Results

$CoCl_2$ did not induce significant changes in beta cell metabolism (NAD(P)H and Rh123), insulin secretion, and survival. Consistent with hypoxia induction, $CoCl_2$ stimulated HIF-1α, PDK-1, and LDHA transcripts and also stimulated VEGF expression and secretion. We observed a modest switch to the less oxidative isoform of COX4 (isoform 1 to 2) and this switch was noted in the glucose-stimulated cytoplasmic NAD(P)H responses. EC morphology and survival were greater in $CoCl_2$ treated islets compared to exogenous VEGF-A in both static (dish) and microfluidic flow culture.

**Data Availability Statement:** The data underlying this study have been uploaded to the Scholars Portal Dataverse: https://doi.org/10.5683/SP2/GCBSSF.

**Funding:** This work was supported by the Natural Sciences and Engineering Research Council (NSERC) of Canada and the Percy Edward Hart and Erwin Edward Hart Professorship from the Faculty of Applied Science and Engineering, University of Toronto. The funders had no role in study design, data collection and analysis, decision to publish, or preparation of the manuscript.

**Competing interests:** The authors have declared that no competing interests exist.

## Conclusions

Hypoxia induction using $CoCl_2$ had a positive effect on islet EC morphology and survival with limited impact on beta-cell metabolism, function, and survival. The EC response appears to be due to endogenous production and secretion of angiogenic factors (e.g. VEGF-A), and mechanistically independent from survival induced by serum albumin.

## Introduction

Pancreatic islet beta-cells sense blood glucose and secrete insulin directly into the blood stream [1]. These functions are supported by a highly tortuous vasculature that places every insulin secreting beta-cell immediately adjacent to a very thin (i.e. lacking a smooth muscle cell layer) and highly fenestrated endothelial cell (EC) [1]. Although pancreatic islets maintain their vasculature immediately after being harvested, it is well established that during *ex vivo* culture EC density decreases to ~50% in the first day and is almost completely gone by the fourth day [2]. The loss of these cells in culture severely limits studying the interaction between beta-cells and ECs. Islet ECs are also involved in the revascularization of transplanted tissue for the treatment for type 1 diabetes yet the EC of donor islets are likely lost due to the common practice of culturing donor islets for 48h to diminish tissue inflammation prior to transplantation [2–4]. It is therefore vital to determine methods to maintain islet-ECs during culture prior to transplantation.

Islets in culture rely on diffusion for media exchange throughout the tissue. Serum albumin is an anti-apoptotic signal for ECs that is sufficiently large and sticky to show restricted diffusion in a tissue. We showed previously that culturing islets in a microfluidic device induces media flow through the tissue, which increases access of serum albumin and subsequently slows the breakdown of vasculature morphology (i.e. islet-EC area and connected length) [5]. However, islet vasculature was only partially maintained by this treatment in comparison to freshly isolated islets. We therefore aimed to determine other treatments to potentially be combined with microfluidic flow to better maintain islet-ECs during culture.

We postulated that one way to slow the loss of ECs in culture was to mimic the normal communication between islet beta-cells and ECs. Epithelial cells, such as beta-cells, communicate with ECs through the balanced secretion of angiogenic and angiostatic factors [6]. Angiogenic factors are paracrine factors that promote and induce growth of new vasculature while angiostatic factors inhibit or suppress the growth of new vasculature. Previous studies attempting to maintain islet-ECs during culture have investigated exogenous growth factors and inhibitors of anti-angiogenic factors, or overexpression of pro-angiogenic genes and silencing of anti-angiogenic genes [7–11]. However, the results are conflicting, and the mechanisms involved have not been elucidated. For example, overexpression of VEGF-A in the developing pancreas leads to a hypervascularized pancreas and ectopic insulin expression [12]. Consistently, knock-out of the anti-angiogenic factor thrombospondin-1 resulted in islet hyperplasia [13,14]. We postulated that ectopic treatments fail to maintain normal EC size and morphology because they inadequately mimic the complex communication (i.e. balance of angiostatic and angiogenic signaling) between beta-cells and ECs. Instead, we aimed to explore treatments that stimulate endogenous expression of angiogenic factors. More specifically, we cultured islets in the hypoxia mimetic cobalt chloride ($CoCl_2$). Using this treatment, we consistently maintained better EC morphology (e.g. area and connected length) and number. We further showed this effect is additive to the response induced by microfluidic flow, consistent with

separate mechanisms. Overall, our data suggest inducing endogenous angiogenic factors by mild hypoxia should be considered as a potential method to slow the demise of ECs during culture and can be done while maintaining islet beta-cell function.

## Materials and methods

### Ethics statement

Animal procedures were approved by the Animal Care Committee of the University Health Network, Toronto, Ontario, Canada in accordance with the policies and guidelines of the Canadian Council on Animal Care (Animal Use Protocol #1531).

### Pancreatic islet isolation and culture

Pancreatic islets were isolated from 10- to 12- week-old male C57BL6 mice using collagenase digestion (Roche) [15,16]. Islets were cultured in RPMI medium 1640 (Sigma-Aldrich) supplemented with 11 mM glucose, 10% FBS, 5 U/ml penicillin-streptomycin, and 20 mM HEPES. Islets treated under static conditions were incubated in non-treated culture dishes (Corning) in a humidified incubator at 37˚C and under 5% $CO_2$. Alternatively, islets cultured in flow were loaded into custom-built microfluidic devices shortly after isolation and incubated in a desk-top incubator [5]. Briefly, the microfluidic device was submerged in 37˚C water bath with flow driven by a syringe pump at a rate of 125 μl/hr (Braintree Scientific Inc.). The media was also submerged in a separate water-bath maintained just above 37˚C to reduce formation of air bubbles in the device. A cap of mineral oil was placed on top of the media to reduce evaporation and pH drift. Islets were cultured for different time points as indicated. 100 μM $CoCl_2$ (Sigma-Aldrich) and recombinant mouse $VEGF_{165}$ (50ng/ml) (eBiosceince) were added to the RPMI 1640 media where indicated.

### Two-photon NAD(P)H imaging

Islets cultured for 24 to 48h as were first equilibrated for approximately 1 hr in imaging media (125 mM NaCl, 5.7 mM KCl, 2.5 mM CaCl2, 1.2 mM MgCl2, 10 mM HEPES, pH 7.4) with 2 mM glucose. NAD(P)H imaging was done as previously described using the 40× 1.3 NA oil immersion objective lens of a LSM710 microscope (Zeiss) and the external non-descanned BiG detector with a custom 385–550 nm band-pass filter [17,18]. The Ti:Saph laser was tuned to 705 nm and attenuated to ~3 mW at the sample (Coherent). Image processing and analysis was done using ImageJ software program version 1.49b. The mean intensity of islets in each image was obtained from 20 to 30 regions of interest (ROIs) that were randomly selected while avoiding saturated pixels found in nonresponsive lipofusion deposits [19]. Five regions distant to the islet were also measured to obtain the average background intensity in each image.

### Insulin secretion

Islets were cultured for 48 hrs in 0 and 100 μM $CoCl_2$ and then handpicked (50 islets per condition) into micro-centrifuge tubes (Eppendorf) for 1 hr of equilibration in 2 mM glucose imaging media at 37˚C. The islets were sequentially stimulated with 2 and 20 mM glucose for 40 min intervals. Supernatant was collected from the tubes before addition of the succeeding stimulation. After collection of the supernatants, total islet insulin content was released by 1% Triton X-100. Fractional total insulin was quantified using a sandwich insulin ELISA assay (Millipore,Billerica, MA).

## Imaging mitochondrial membrane potential

Islets were incubated in 2 mM glucose-imaging buffer in the presence of Rh123 (10 mg/ml; 30 min at 37°C) and subsequently imaged using a Zeiss LSM710 confocal (514-nm laser line, 525–655-nm bandpass filter, pinhole size of 1.73 AU, and a pixel dwell time of 12.6 ms). Image analysis was done as described in NAD(P)H imaging.

## VEGF-A ELISA

Islets supernatant was collected after the indicated treatments, and total VEGF-A protein was released from the tissue by 1% Triton-X 100. VEGF-A protein levels were quantified using a sandwich Human VEGF Quantikine ELISA kit (R and D Systems).

## Reverse transcription-PCR and quantitative real-time PCR (qRT-PCR)

Total RNA was extracted from cells and tissue using Trizol (Invitrogen) and collected using the Wizard®SV Gel and PCR clean-up System (Promega) following the manufactures' protocol. RNA concentration and purity were checked using a NanoDrop 2000 Biophotometer (Thermo Fisher Scientific Inc.)

One microgram of total RNA was used for first-strand DNA synthesis using iScript cDNA Synthesis system (BioRad). The master mix (total volume 20 μl) was composed of 10 ng mouse islet cDNA, 10X Taq reaction buffer, 1 μl of each primer (10 mM final concentration), 50 μM dNTPs, 0.5 μl Taq DNA polymerase (New England Biolabs) and nuclease free water. An MJ Research PTC-200 Peltier Thermal Cycler was used with the following cycling parameters: 94°C for 30 s, 52°C for 1 min, and 72°C for 1 min (30 cycles). The housekeeping gene β-actin was amplified as a positive control using identical cycling parameters and all reactions included nuclease-free water as a negative control.

The following primers were designed and used: *VEGF-A*, (5'-GGCCATAGAAGTTTGGC AAG-3' and 5' CCTCTCCCGTGTACAGCTTC-3') *HIF-1α*, (5'-GGTTCCAGCAGACCCA GTTA-3' and 5'- AGGCTCCTTGGATGAGCTTT-3'); *PDK-1* (5'-GGCGGCTTTGTGATT TGTAT-3' and 5'-ACCTGAATCGGGGATAAAC-3'), *LDHA* (5'- TCCAGCAAAGACTA CTGTGT-3' and 5'- GAAGATGTTCACGTTTCGCT-3'), *COX4-1* (5'- TGGCAAGAGA GCCATTTCTA-3' and 5'- TCGTTAAACTGGATGCGGTA-3') & *COX4-2* (5'- GGCCC TGAAGGAGAAAGAGA-3' and 5'-ATGAAGAAGAAGACGCAGCC-3'), *β-actin* (5'-ATCG AGCTCATCCCAT CACCATCTTCCAGG-3' and 5'-ACATCTAGAGCCATCACGCCACAGT TTCCC-3').

The PCR products were visualized by electrophoresis on a 2% agarose gel in 0.5× TBE buffer after staining with 0.5 μg/ml ethidium bromide. The UV-illuminated gels were photographed.

*VEGF-A* levels in the mouse islets were measured by quantitative PCR using SYBR® Green predesigned *VEGF-A* primers (Sigma-aldrich). Following the manufacturer's protocol: 20ng of reverse-transcribed cDNA was subjected to quantitative real time PCR using SYBR® Green JumpStart™ Taq ReadyMix™. Each reaction comprised 1x qPCR mix, 0.2μM forward and reverse primer, 20ng cDNA (or distilled H₂O), and PCR grade water to a final volume of 20μL.

## Immunofluorescent detection

Islets were sequentially fixed (2% PFA, PBS, 1 hr), blocked (PBS, 0.1% TritonX-100, 4°C, 10% Normal Goat Serum, 4 hr), incubated with primary antibody (PBS,0.1% TritonX-100, 4°C, 1% Normal Goat Serum, 1:100 dilution rat anti-mouse PECAM-1, 3 hr), rinsed (PBS, 0.1% TritonX-100, 4°C, 10% Normal Goat Serum, 1 hr) and incubated with secondary antibody (1:500

goat anti-rat Alexa 633 in PBS, 0.1% TritonX-100, 4˚C, 10% Normal Goat Serum, 3 hr). Islets were imaged using a Zeiss LSM 710 microscope. Images were analyzed using ImageJ version 1.49b. EC fractional area was assessed by measuring the area of PECAM-1 positive labeling relative to the total area of the islet. The EC connected length was measured using Simple Neurite Tracer throughout all the images collected. PECAM-1 positive regions were traced through the image stacks. The summation of the traces was normalized to the circumference of the islets. The average of individual islets was used for statistical analysis.

## Microfluidic device fabrication

The microfluidic devices were made using elastomer polydimethylsiloxane (PDMS) (Dow-Corning) as described previously [20,21]. The PDMS was cured and holes were inserted into the mold. The cured PDMS was permanently bonded to 24 mm x 50 mm number one glass coverslips (VWR Scientific) by oxygen plasma treatment (Harrick Scientific, Ossining, NY). Tygon tubing was inserted into the inlet and outlet pores of the PDMS to allow for loading of the islets.

## Statistical analysis

Data are presented as mean ± SEM of at least three independent experiments. Statistical significance was assessed by either two-tailed, unpaired student's t-test or one-way ANOVA with post-hoc Bonferoni test as indicated.

## Results

### CoCl$_2$ induces a hypoxic response in islets, without significantly impairing glucose-stimulated mitochondrial metabolism, insulin secretion, or survival

CoCl$_2$ is commonly used at a concentration of 100 μM to mimic hypoxia in cultured cells and tissues [22–24]. To determine if this treatment adversely affects islet glucose responses, we examined glucose-stimulated metabolism and insulin secretion (Fig 1A–1C). Islets cultured in 100 μM CoCl$_2$ for 24hr showed similar glucose-stimulated NAD(P)H response, mitochondrial membrane potential, and insulin secretion compared to control (0 μM CoCl$_2$) islets. These data suggest limited effect of 100 μM CoCl$_2$ on islet stimulus coupling. To determine the effect of CoCl$_2$ on islet cell survival, we explored apoptosis of insulin positive cells using cleaved caspase-3 staining (Fig 1D). These data show negligible apoptosis compared to control, suggesting that hypoxia induced by 100 μM CoCl$_2$ is relatively mild and nontoxic to islets. Lastly, to confirm the induction of hypoxia in islets, we measured hypoxia-inducible factor-1α (*HIF-1α*) mRNA expression by qPCR (Fig 1E). These data show a nearly 8-fold increase in *HIF-1α* expression consistent with CoCl$_2$-induced hypoxia. Overall, these data show that 100 μM CoCl$_2$ induces a hypoxic response in islets, but does not impair beta-cell mitochondrial metabolism, insulin secretion, or survival. We therefore used 100 μM CoCl$_2$ to induce hypoxia in islets for the remainder of the study.

### CoCl$_2$ maintains beta-cell glucose metabolism by enhancing glycolysis while preserving mitochondrial metabolism

It is well established that hypoxia can induce a shift in glucose-stimulated metabolism lowering oxidative phosphorylation to mainly rely on glycolysis [14–16, 25]. To determine the effect of CoCl$_2$ on the underlying metabolism of islet beta-cells, we assessed metabolic activity associated with cytoplasmic (i.e. glycolytic) and mitochondrial (i.e. OXPHOS)

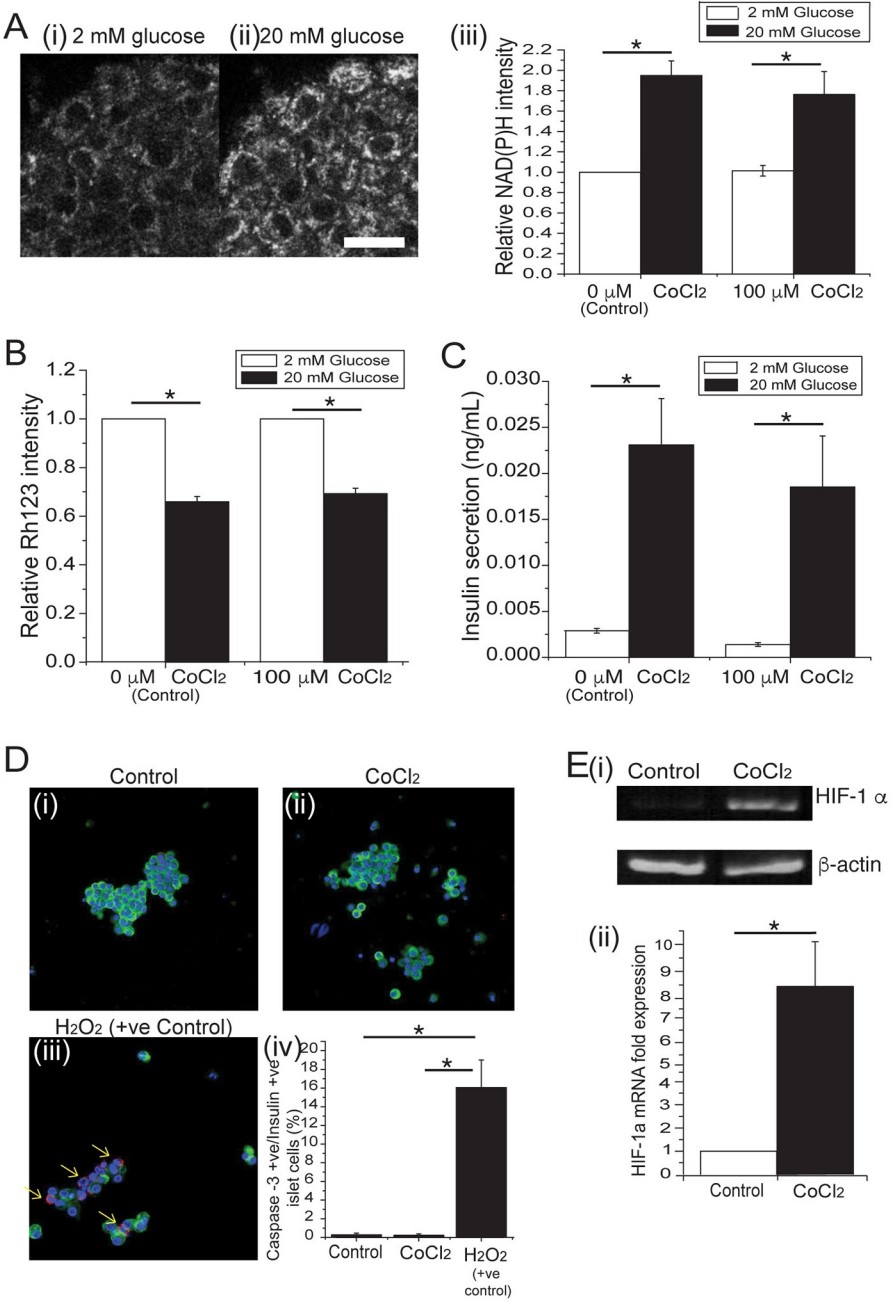

**Fig 1. Induction of moderate hypoxia in isolated islets with CoCl₂. (A)** A representative two-photon NAD(P)H image of an isolated pancreatic islet in **(i)** 2mM and **(ii)** 20 mM glucose (Scale bar = 40μm). **(A)(iii)** Relative NAD(P)H intensities of islets cultured in the absence (0 μM) and presence (100 μM) of CoCl₂ (n = 3 mice on separate days). **(B)** Relative Rh123 intensity of islets in response to 2mM (white bars) and 20mM (black bars) glucose (n = 5 mice on separate days). **(C)** Insulin secretion (ng/mL) from islets in response to 2 mM (white bars) and 20 mM (black bars) glucose (n = 4 mice on separate days). **(D)(i-iii)** Representative images of islets dispersed to single cells after being cultured in the absence (0 μM) and presence (100 μM) CoCl₂. Islets dispersed into single cells were stained for insulin (green), caspase-3 (red) and DAPI (blue). Arrows show caspase-3 positive cells in H₂O₂-treated cells (H₂0₂: +ve control). **D(iv)** Images were subsequently quantified to determine the percent of Caspase-3 +ve / insulin +ve cells (n = 3 mice on separate days). **(E)(i)** Image of RT-PCR and **(ii)** qPCR of HIF-1α transcript expression of islets (n = 5 mice on separate days). Error bars are ± SEM. * p < 0.05. All images were identically Gaussian filtered and stretched for visualization.

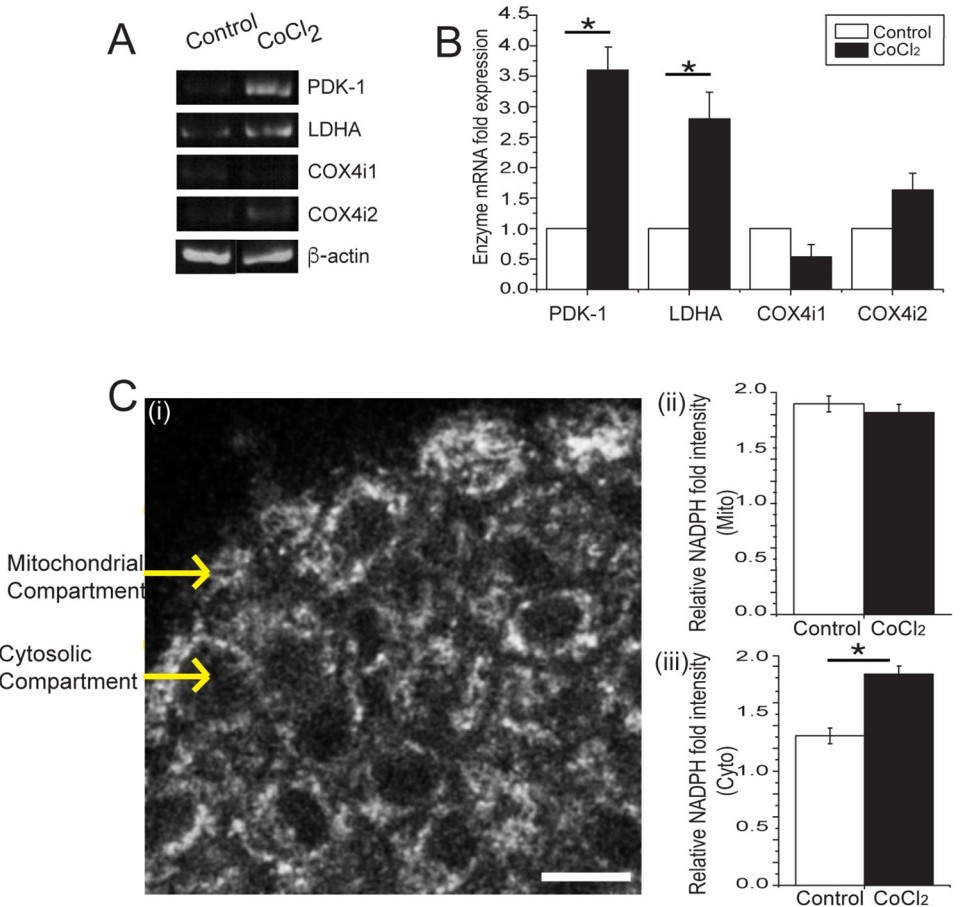

**Fig 2. Effect of CoCl₂ on islet beta-cells.** Islets isolated from C57Bl6 mice were cultured in the absence (0 μM) and presence (100 μM) CoCl₂ for 24 hrs. **(A)** RNA was extracted from islets and RT-PCR was run to determine the presence of metabolic enzymes (PDK-1, LDHA, COX4i-1 & COX4i-2) compared to the house-keeping gene β-actin (n = 10 mice on separate days). **(B)** Bar graph shows the quantification of RNA results from **(A)**. mRNA fold expression of metabolic enzymes. White bars represent control (0μM CoCl₂) islets; black bars represent CoCl₂ (100 μM) treated islets. **(C)(i)** High-resolution NADP(H) image of an islet using two-photon microscopy. Yellow arrows point to the mitochondrial and cytoplasmic compartments (scale bar = 15μm). **(C)(ii)** Bar graphs show the relative NAD(P)H intensity in the mitochondrial compartment (Bar graph at top) and **(C)(iii)** in the cytoplasmic compartment (Bar graph at bottom). White bars represent control (0μM CoCl₂) islets; black bars represent CoCl₂ (100 μM) treated islets (n = 10 mice on separate days). Error bars are ± SEM. * p < 0.05. All images were identically Gaussian filtered and stretched for visualization purposes.

metabolism (Fig 2). Firstly, we investigated transcript expression of metabolic enzymes classically induced by hypoxia. Islets treated with 100 μM CoCl₂ showed a significant increase in glycolytic enzymes (*PDK-1* and *LDH-A*) and an isoform change from mitochondrial *COX4-1* to *COX4-2* (Fig 2A and 2B). These data are consistent with enhanced glycolytic metabolism and diminished oxidative phosphorylation. Second, we used the high spatial resolution of two-photon NAD(P)H imaging to independently measure cytoplasmic and mitochondrial NAD(P)H responses (Fig 2C). Islets treated with CoCl₂ showed no significant difference in glucose-stimulated mitochondrial (Mito) NAD(P)H responses compared to control (Fig 2C(ii)). However, we observed a significantly larger cytoplasmic (Cyto) response in CoCl₂ treated islets (Fig 2C(iii)). These data are consistent with greater glycolytic metabolism in response to hypoxia. Altogether, these data suggest that CoCl₂

maintains beta-cell glucose metabolism by enhancing glycolysis while preserving mitochondrial metabolism.

## CoCl$_2$ induces endogenous VEGF-A expression that leads to increased vessel length, and EC-area, -number and -survival

The effect of hypoxia on the expression and secretion of growth factors, such as VEGF, is well documented [6–11]. We postulated that induction of these factors in islets could potentiate survival of the ECs in culture. To explore the effect of endogenous angiogenic factors on islet ECs, we investigated the effect of the hypoxia mimetic CoCl$_2$ on islet angiogenic factor expression and vessel morphology (Fig 3). First, we examined CoCl$_2$-induced VEGF-A transcript and protein expression (Fig 3A(i) and 3A(ii)). These data show significant induction of *VEGF-A* mRNA (24hrs and 48 hrs) and protein (48 hrs) expression in islets in response to CoCl$_2$. Next, we investigated the effect of CoCl$_2$ and exogenous VEGF-A addition on islet EC branching and morphology throughout islets (Fig 3B and 3C). These data show a progressive loss in EC area and connected length in all of the islet treatment groups consistent with a progressive loss in EC number and vessel morphology. However, islets cultured in CoCl$_2$ maintained significantly more EC area and connected length compared to control and VEGF-A treated islets at 48hrs and 72 hrs (Fig 3B and 3C). These data suggest that hypoxia maintains EC number and morphology better than untreated islets, and better than exogenous addition of VEGF-A. Lastly, to determine the effect of CoCl$_2$ on islet EC viability, we examined whether EC number and apoptosis correlated with a decline in vessel morphology (Fig 3D and 3E). After treatment with CoCl$_2$, islets were labeled with PECAM-1 and DAPI to count the number of living EC. These data show more PECAM-1 positive ECs present in CoCl$_2$–treated islets at 24 and 48 hrs compared to control islets (Fig 3D(i)–3D(iv)). When looking at islets labeled with PECAM-1, DAPI and caspase-3, we observed a significantly lower number of caspase-3 positive EC in CoCl$_2$-treated islets at 24 and 48 hrs compared to control islets (Fig 3E(i)–3E (v)). Taken together, these data confirm that the hypoxia mimetic CoCl$_2$ induces VEGF-A expression endogenously (via *HIF-1α*) leading to enhanced vessel length and increased EC area, number and survival rate.

## CoCl$_2$ treatment and microfluidic flow have additive effects on EC morphology suggesting they work by independent mechanisms

We previously showed that culturing islets in microfluidic devices improves EC area and connected length [5]. This effect was due to enhanced media transfer resulting in greater tissue penetration of serum albumin to positively affect EC survival. To determine the combined effect of microfluidic flow and hypoxia on islet EC morphology, we examined PECAM-1 positive ECs in islets cultured in static and microfluidic flowing media in response to CoCl$_2$ and exogenous VEGF-A (Fig 4). We used a microfluidic device to hold islets stationary during culture (Fig 4A) [5]. This device uses a drop in the channel height to trap the islets with media flowing both around and through the tissue (Fig 4A(i)). Media flow through the islet is induced by the high fluidic resistance of the microfluidic channels and a resulting pressure drop across the tissue. Multiple microfluidic channels were run on a single chip with independent entrance and exit tubing, which allowed us to simultaneously culture islets with multiple treatments (Fig 4A and 4A(ii)). We cultured islets in this device with 100 μM CoCl$_2$ and 100 ng/ml VEGF-A for 48hrs. Consistent with our previous study, islets cultured in microfluidic flow maintain significantly more EC area and connected length (Fig 4B–4D, **Control**). CoCl$_2$ treatment again showed a significant increase in EC area and connected length respectively, compared to control (Fig 4C and 4D, **CoCl$_2$**). Importantly, microfluidic flow had an additive

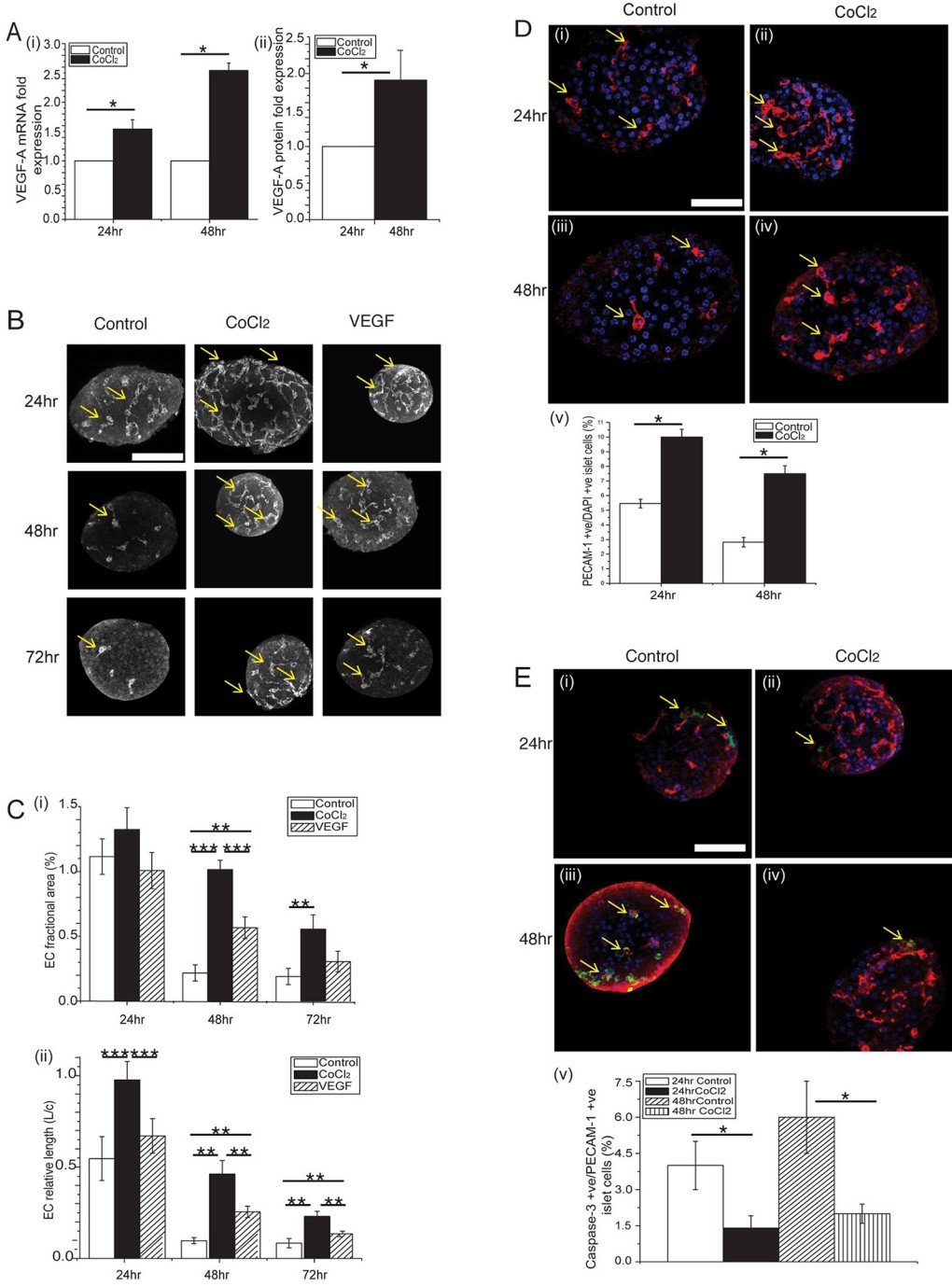

**Fig 3. Effect of CoCl₂ on islet VEGF expression and EC-morphology and -number. (A)(i)** RNA was extracted from islets cultured in varying CoCl₂ for up to 48hrs. Bar graph represents VEGF-A mRNA levels in control (0μM CoCl₂) islets (white bars) compared to 100 μM CoCl₂ treated islets (black bars) (n = 5 mice on separate days). **(A)(ii)** Bar graph shows VEGF-A protein expression of islets cultured over 48hrs (n = 10 mice on separate days). Error bars are ± SEM; *p<0.05. Statistical significance was determined by two-tail student's t-test. **(B)** Image panel of control (0μM CoCl₂); 100 μM CoCl₂ and 100 ng/ml VEGF-A treated islets cultured for up to 72hrs, fixed and immunoflourescently labeled with Alexa Fluor 633 for PECAM-1 (shown in gray); yellow arrows point to PECAM-1 positive cells (scale bar = 50μm). **(C)(i)** Bar graph showing the EC fractional areas of differently treated islets cultured over 24hrs, 48hrs and 72hrs. (n = 15 mice on separate days). **(C)(ii)** Bar graph shows the EC relative length of islets cultured over 24hrs, 48hrs and 72hrs (n = 15 mice on separate days). Error bars are ± SEM; **, p<0.01; ***, p<0.001. Statistical significance was determined by one-way ANOVA with post-hoc Bonferroni test. **(D)(i-iv)** Image panel of control (0μM CoCl₂) and 100 μM CoCl₂ treated islets

cultured for 24hrs and 48hrs, fixed and immunoflourescently labeled for PECAM-1 (shown in red) and DAPI (shown in blue); yellow arrows point to PECAM-1 positive and DAPI positive cells (scale bar = 50μm). Bar graph at bottom of **(D)** **(v)** shows the percentage of PECAM-1 positive and DAPI positive ECs in islets cultured in the presence and absence of $CoCl_2$ (n = 5 mice on separate days). **(E)(i-iv)** Image panel of islets cultured for 24 and 48hrs, fixed and immunoflourescently labeled for PECAM-1 (red), cleaved caspase-3 (green), and DAPI (blue); yellow arrows point to triple positive cells (scale bar = 50 μm). **(E)(v)** Bar graph at bottom shows the percentage of PECAM-1 positive + DAPI positive + cleave-caspase-3 positive ECs in islets cultured in varying $CoCl_2$. White bar represents control 24 hr islets; black bar represents $CoCl_2$ 24 hr islets; diagonal shaded bar represents control 48 hr islets and vertical shaded bar represents $CoCl_2$ 48hr islets (n = 5 mice on separate days). Error bars are ± SEM; *p<0.05. Statistical significance was determined by two-tail student's t-test.

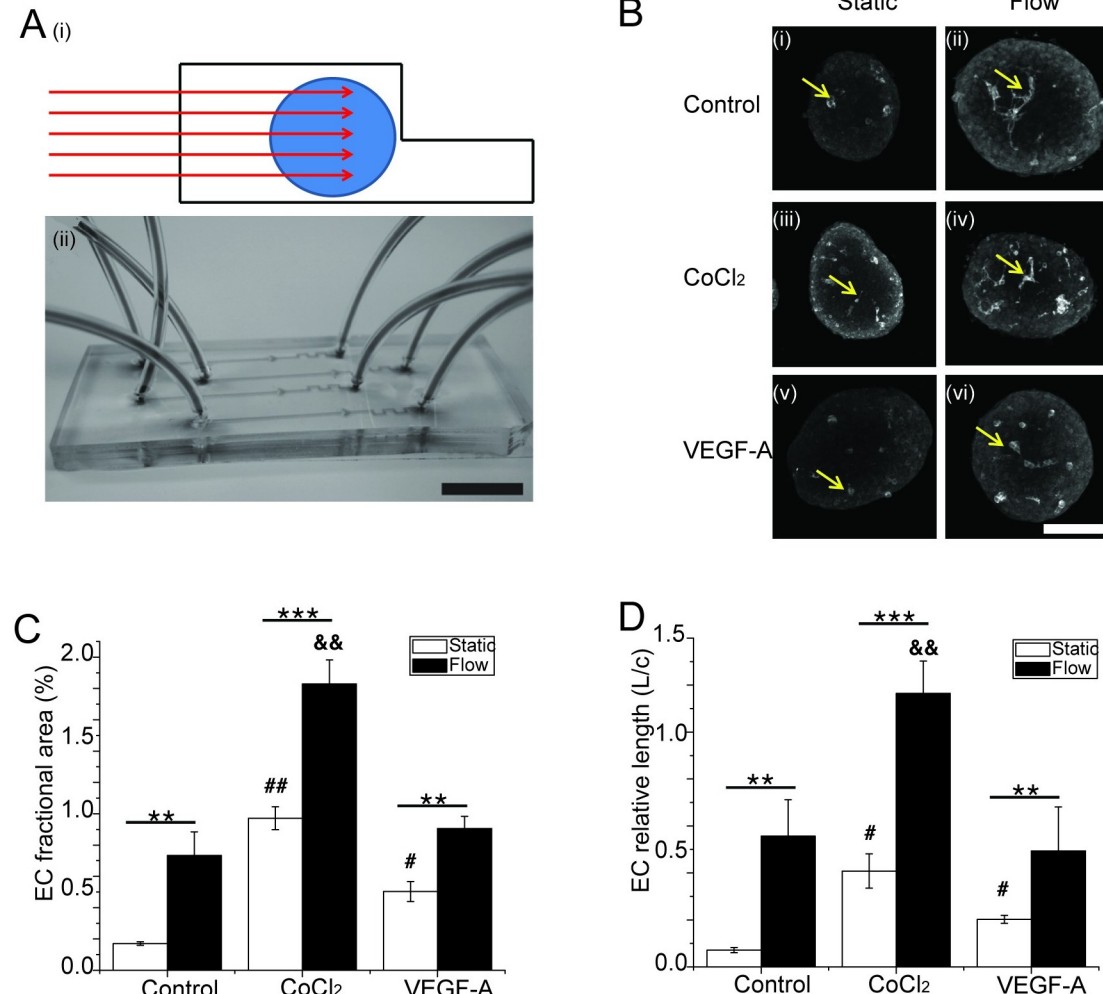

**Fig 4. Effect of $CoCl_2$, microfluidic flow, and exogenous VEGF-A on EC morphology. (A) (i)** Schematic and **(ii)** image of a microfluidic device used to culture pancreatic islets in flowing media. This device holds islets in a dam-wall to subsequently induce the flow of media both around and through the tissue (direction of flow shown in (A)(i) schematic). The device was filled with food coloring to illustrate the inlet and outlet tubing and 100 μm tall channels. Each device has 4 channels, which allowed simultaneous culture of the islets in control (0μM $CoCl_2$), 100 μM $CoCl_2$, and 100 ng/ml VEGF-A (scale bar = 10mm). **(B)(i-vi)** Representative extended focus projection images of islets cultured under static and flow conditions and treated with the previously mentioned factors. The islets were subsequently immunofluorescently labeled for PECAM-1 to illustrate the ECs (shown by arrows) (scale bar = 50 μm). **(C & D)** The image stacks from **(B)** were quantified to determine the EC fractional area (C) and EC relative length (D) for each islet; (n = 6 mice on separate days). Statistical significance was determined by one-way ANOVA with post-hoc Bonferroni test. Error bars are ± SEM; **, p<0.01; ***, p<0.001. #, p<0.01 compared to static control; ##, p<0.001 compared to static control. &&, p<0.001 compared to flow control.

effect on the response induced by $CoCl_2$, that was also greater than the combined effect of flow with VEGF-A treatment. The inability of the combined VEGF-A and flow treatment to induce a similar response to flow and $CoCl_2$, suggests that VEGF-A is not the only factor induced by hypoxia and thus not entirely responsible for the observed effect on vessel morphology. Altogether, our data show that $CoCl_2$ and microfluidic flow have additive effects on EC morphology suggesting they work by independent mechanisms.

## Discussion

Pancreatic islets are heavily vascularized *in vivo* to facilitate glucose sensing and insulin secretion into the blood stream. This vascularization is presumably achieved through tight communication between insulin secreting beta-cells and ECs. However, this communication is somehow perturbed in culture resulting in a slow loss of vascularity and EC number over a period of days. We previously showed that culturing islets in flowing media using a microfluidic device slowed the loss of EC morphology, and that this response was due to enhanced penetration of the antiapoptotic factor serum albumin into the tissue. In the present study, we endogenously manipulated the morphology and survival of islet ECs in culture using the hypoxia mimetic $CoCl_2$. We were initially concerned that $CoCl_2$ would negatively impact beta-cell metabolism and function, yet our data show only a moderate shift to glycolytic metabolism, and limited impact on insulin secretion and beta-cell survival. Consistent with induction of hypoxia, $CoCl_2$ upregulated both *HIF-1α* expression and VEGF-A expression and secretion, and ultimately slowed the loss of EC morphology and number. Importantly, the effect of hypoxia was additive to flowing media suggesting they both work by independent mechanisms. Furthermore, combining $CoCl_2$ and flow maintained EC morphology better than exogenous VEGF-A and flow suggesting hypoxia induced more than expression of this single factor. Overall these data indicate that hypoxia and laminar flow in a microfluidic device should be considered for long-term culture and maintenance of pancreatic islets *ex vivo*.

Beta-cells respond to glucose through metabolic generation of NAD(P)H resulting in a cascade of closure of ATP sensitive potassium channels ($K_{ATP}$), membrane depolarization, $Ca^{2+}$ influx and insulin secretion [26]. We were therefore initially concerned that hypoxia induction might impact insulin secretion. We examined the glucose-stimulated metabolic responses of islets using two-photon excitation of NAD(P)H autofluorescence and Rh123 imaging of mitochondrial membrane potential. We found that 100 μM $CoCl_2$ did not significantly perturb either glucose-stimulated response. Consistently, we found normal glucose-stimulated insulin secretion suggesting the hypoxia mimetic has moderate impact on beta-cell function. Yet a previous study by Sato *et al.* found that constitutive activation of HIF-1α inhibited GSIS [27]. This varying result may be due to the differences in the mode and intensity of activation: they used varying oxygen tensions to stimulate HIF-1α while we worked with 100 μM $CoCl_2$ due to early work showing higher concentrations (250 and 500 μM) blunted the metabolic response (data not shown). Importantly, we also showed no increase in beta-cell apoptosis, leading us to conclude the hypoxia mimetic at this concentration has limited impact on beta-cell metabolism, function and survival.

Previous work investigated stimulating islets using exogenous growth factors, inhibitors of anti-angiogenic factors, or overexpression of pro-angiogenic genes and silencing of anti-angiogenic genes [7–11,13,14]. Overexpression of VEGF-A and Ang-1 separately and together in islets was found to improve vascular density of islet grafts, increase blood flow and insulin content [7–9]. Gene silencing of angiostatic factor thrombospondin-1 has also improved vascularisation post transplantation [10]. However, despite the positive results reported, other studies have found that islets treated with exogenous VEGF or FGF-2 had no

significant impact on revascularization compared to controls [11]. Overexpression of VEGF has also been linked to hyperpermeability in the vasculature causing it to be leaky [13]. Additionally, TSP-1 knock-out mice have been reported to develop pancreatic islet hyperplasia with increased vascular density [14]. In contrast, our study used the hypoxia-mimetic $CoCl_2$ to stimulate endogenous angiogenic factors. Consistent with hypoxia induction, we showed that $CoCl_2$ induced *HIF-1α* a master regulator of angiogenic factor expression and metabolism in different cell types and tissues [16, 25, 28]. We also found increased VEGF-A production along with increased glycolytic enzyme levels (*PDK-1* and *LDHA*), while switching mitochondrial *COX4* enzyme isoforms to enhance metabolic efficiency. NAD(P)H imaging provided sufficient spatial resolution to determine the effect in mitochondrial and cytoplasmic compartments [19]. On further investigation of the mitochondrial and cytoplasmic NAD(P)H levels (representative of glycolytic metabolism), the results corresponded to the metabolic enzyme profile we observed. Taken together, these results suggest that $CoCl_2$ stimulates hypoxia while still maintaining islet beta-cell function and metabolic efficiency via *HIF-1α* and its associated genes.

We previously showed that laminar flow in a microfluidic device led to enhanced maintenance of islet-ECs [5]. We therefore compared $CoCl_2$ and flow to islets in flow alone and with exogenous VEGF-A. We found enhanced islet-EC area and connected length in all treatments under flow conditions. However, the response was significantly more significant under $CoCl_2$ treatment compared to control and VEGF-A treatment suggesting the impact of flow and hypoxia are additive. A possible mechanism is hypoxia-induced production of various angiogenic factors acting on EC junctional proteins such as PECAM-1 or VE-cadherin. Previous studies showed that angiogenic factors such as VEGF, FGFs and Angiopoietins lead to either break down or maintenance of VE-cadherin junctions between ECs [29–33]. Future studies will aim to investigate the relationship between $CoCl_2$ induced angiogenic factor production and junctional proteins such as VE-cadherin. Additionally, since VEGF is a chemotactic agent for ECs, we posit that giving it exogenously to islets could result in a gradient into the tissue to induce vessel breakdown and migration of ECs out of the islet. However, we anticipate that such a gradient would be mitigated by flow, thus we believe the main difference between $CoCl_2$ and exogenous VEGF is that $CoCl_2$ stimulates a mixture of factors from within the islet itself; thereby better mimicking the complexity and geometry of normal communication between beta-cells and ECs.

## Acknowledgments

The authors declare that they have no conflicts of interest with the contents of this article.

## Author Contributions

**Conceptualization:** Krishana S. Sankar, Jonathan V. Rocheleau.

**Data curation:** Krishana S. Sankar, Svetlana M. Altamentova.

**Formal analysis:** Krishana S. Sankar.

**Funding acquisition:** Jonathan V. Rocheleau.

**Supervision:** Jonathan V. Rocheleau.

**Writing – original draft:** Krishana S. Sankar.

**Writing – review & editing:** Krishana S. Sankar, Jonathan V. Rocheleau.

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
