## [Decision Letter · Decision Letter 0]

12 Jul 2019

PONE-D-19-16565

Hypoxia Induction in Cultured Pancreatic Islets Enhances Endothelial Cell Morphology and Survival while Maintaining Beta-cell Function

PLOS ONE

Dear Dr. Rocheleau,

Thank you for submitting your manuscript to PLOS ONE. After careful consideration, we feel that it has merit but does not fully meet PLOS ONE’s publication criteria as it currently stands. Therefore, we invite you to submit a revised version of the manuscript that addresses the points raised during the review process.

We would appreciate receiving your revised manuscript by Aug 26 2019 11:59PM. To enhance the reproducibility of your results, we recommend that if applicable you deposit your laboratory protocols in protocols.io, where a protocol can be assigned its own identifier (DOI) such that it can be cited independently in the future. For instructions see: http://journals.plos.org/plosone/s/submission-guidelines#loc-laboratory-protocols

We look forward to receiving your revised manuscript.

Kind regards,

Feng Zhao

Academic Editor

PLOS ONE

Journal Requirements:

2. For ease of reproducibility we would recommend that you amend your methods section and provide further details of how the pancreatic islets were isolated from 10- to 12- week-old male C57BL6 mice.

**Comments to the Author**

1. Is the manuscript technically sound, and do the data support the conclusions?

Reviewer #1: Yes

Reviewer #2: Yes

2. Has the statistical analysis been performed appropriately and rigorously? 

Reviewer #1: No

Reviewer #2: No

3. Have the authors made all data underlying the findings in their manuscript fully available?

Reviewer #1: Yes

Reviewer #2: Yes

4. Is the manuscript presented in an intelligible fashion and written in standard English?

Reviewer #1: Yes

Reviewer #2: No

5. Review Comments to the Author

Reviewer #1: Authors continued their previous study on how to maintain the pro- and anti-angiogenic balance to preserve vascular structures in the cultured pancreatic islets, which is critical for maintaining tissue viability and revascularization for transplantation for diabetes treatment. By using cultured islets collected from mice as the model, it is proposed that CoCl2 treatment, which is a hypoxia mimic, causes slower loss of ECs, while islet viability and insulin secretion functionality were not significantly affected. They also showed the correlation between the increased HIF-α/VEGF-A resulted from CoCl2 treatment and the increased ECs survival. And this effect can be augmented by using flowing culture. The topic is interesting and the flow-through of this manuscript is very clear. Before final publication it is suggested to do the following adjustment.

1. Some terms are not properly defined. For example, EC is mentioned in Intro for the first time lacks explanation. What is +ve control (In figure 1 caption)? Sometimes HIF1α was used; sometimes HIF-1α or HIF-1 was used. Please double check and be consistent.

2. Figure captions, especially for Fig 3, are too long and distracting. The data are already self-explanatory. There is no need to repeat the experimental conditions. Meanwhile, some essential info is missing. Please add explanation on different fluorescent colors either in the figure or in the captions.

3. This concern is related to the sentence in the discussion section “we manipulated the communication between beta-cells and ECs using the hypoxia mimetic CoCl2 “.

Since this study used isolated islet, which contains multiple cell type, can increased of H1F-1α and VEGF-A be attributed to other cells? Please explain why authors only credit the communication between beta-cell and ECs?

4. The ANOVA tests performed according to the presented results are not proper. ANOVA test tells whether differences between groups are statistically significant. Then post hoc tests (such as Tukey or LSD) should be performed to detect where the difference(s) are. The result should include the suggested grouping, not individually comparing with control, which could easily makes Type 1 error. Please report the proper result unless there is a specific reason for using t-test after ANOVA. Thus an alert should be given in the paper, too.

Reviewer #2: This manuscript confirmed that incubation isolated islets with hypoxia mimic cobalt chloride (CoCl2) induced endothelial cell morphology and cell survival. This hypoxia induction did not impair the beta-cell function based on the data on insulin secretion and beta-cell survival. Importantly, CoCl2 increased expression of HIF-1α and VEGF-A. The authors also showed that culturing islets in a microfluidic device combined with CoCl2 enhanced endothelial cell morphology and survival.

Some concerns:

1. In both “Introduction” and “Discussion” part, the authors stated “Previous studies attempting to maintain islet-ECs during culture have investigated exogenous growth factors and inhibitors of antiangiogenic factors, or overexpression of pro-angiogenic genes and silencing of anti-angiogenic genes”. More detailed and updated research information needs to be described and discussed.

2. In all the figures, the multiple significant differences between groups on a bar graph were marked not clear. The resolution of microscopy images can be improved.

3. In Figure 1, CoCl2 significantly increased HIF-1α expression (8-fold), but had no effect on insulin secretion. However, previous data (J Biol Chem 286: 12524–12532) reported that a constitutive activation of HIF-1α inhibit GSIS. This should be addressed.

6. PLOS authors have the option to publish the peer review history of their article (what does this mean?). If published, this will include your full peer review and any attached files.

---

## [Author Response · Author response to Decision Letter 0]

20 Aug 2019

We would like to thank the reviewers for their time and effort in making this manuscript better. Overall, the reviewers indicated that the “topic is interesting” and that the “manuscript is very clear”. However, each reviewer had specific concerns that needed to be addressed prior to publication. Our responses to these concerns are shown below.

Reviewer #1:

Authors continued their previous study on how to maintain the pro- and anti-angiogenic balance to preserve vascular structures in the cultured pancreatic islets, which is critical for maintaining tissue viability and revascularization for transplantation for diabetes treatment. By using cultured islets collected from mice as the model, it is proposed that CoCl2 treatment, which is a hypoxia mimic, causes slower loss of ECs, while islet viability and insulin secretion functionality were not significantly affected. They also showed the correlation between the increased HIF-α/VEGF-A resulted from CoCl2 treatment and the increased ECs survival. And this effect can be augmented by using flowing culture. The topic is interesting and the flow-through of this manuscript is very clear. Before final publication it is suggested to do the following adjustment.

Concerns

1. Some terms are not properly defined. For example, EC is mentioned in Intro for the first time lacks explanation. What is +ve control (In figure 1 caption)? Sometimes HIF1α was used; sometimes HIF-1α or HIF-1 was used. Please double check and be consistent.

Thank you for pointing out these oversights. Please see the following corrections in the manuscript: (1) endothelial cell (EC) is now defined in the first instance of the introduction (pg. 3); (2) H202 treated cells are now clearly defined as the +ve control in figure 1 caption (pg. 11); and (3) we now consistently use HIF-1α throughout (pgs. 2,8,10,11,14,18,19). 

2. Figure captions, especially for Fig 3, are too long and distracting. The data are already self-explanatory. There is no need to repeat the experimental conditions. Meanwhile, some essential info is missing. Please add explanation on different fluorescent colors either in the figure or in the captions.

To address this concern, we edited all of the figure captions. We have also included information on the different fluorescent colours used in the figure captions (pgs. 11,14, 15, 17).

3. This concern is related to the sentence in the discussion section “we manipulated the communication between beta-cells and ECs using the hypoxia mimetic CoCl2 “.

Since this study used isolated islet, which contains multiple cell type, can increased of H1F-1α and VEGF-A be attributed to other cells? Please explain why authors only credit the communication between beta-cell and ECs?

We completely agree that this sentence was overly presumptive in assigning the mechanism solely to beta-cell secretion of angiogenic factors. We attribute this presumption to the fact that islets comprise 85% beta-cells but agree that our data does not specifically negate the impact of other islet cells (e.g. alpha, delta, and pp cells). To address this concern, we have modified the sentence to state: “In the present study, we endogenously manipulated the morphology and survival of islet ECs in culture using the hypoxia mimetic CoCl2.”

4. The ANOVA tests performed according to the presented results are not proper. ANOVA test tells whether differences between groups are statistically significant. Then post hoc tests (such as Tukey or LSD) should be performed to detect where the difference(s) are. The result should include the suggested grouping, not individually comparing with control, which could easily makes Type 1 error. Please report the proper result unless there is a specific reason for using t-test after ANOVA. Thus an alert should be given in the paper, too.

Thank you for bringing this to our attention. We agree that using an appropriate post hoc test is important to avoid type 1 errors. We revisited our analyses and used Bonferroni post hoc test to compare between all groups. Please see our modifications in the manuscript (pgs. 15 and 17) and to the figures throughout the manuscript.

Reviewer #2:

This manuscript confirmed that incubation isolated islets with hypoxia mimic cobalt chloride (CoCl2) induced endothelial cell morphology and cell survival. This hypoxia induction did not impair the beta-cell function based on the data on insulin secretion and beta-cell survival. Importantly, CoCl2 increased expression of HIF-1α and VEGF-A. The authors also showed that culturing islets in a microfluidic device combined with CoCl2 enhanced endothelial cell morphology and survival.

Some concerns:

1. In both “Introduction” and “Discussion” part, the authors stated “Previous studies attempting to maintain islet-ECs during culture have investigated exogenous growth factors and inhibitors of antiangiogenic factors, or overexpression of pro-angiogenic genes and silencing of anti-angiogenic genes”. More detailed and updated research information needs to be described and discussed. 

We agree and have now expanded on these details in the discussion (pg. 19).

2. In all the figures, the multiple significant differences between groups on a bar graph were marked not clear. The resolution of microscopy images can be improved.

Thank you for pointing out this oversight. We have now added bars indicating the significance between groups to clearly distinguish comparison and significance in all figures. We agree that on a journal page the images will be very small. However, the actual resolution is close to the images collected and if needed, one can digitally zoom in to look at the details more closely.

3. In Figure 1, CoCl2 significantly increased HIF-1α expression (8-fold), but had no effect on insulin secretion. However, previous data (J Biol Chem 286: 12524–12532) reported that a constitutive activation of HIF-1α inhibit GSIS. This should be addressed.

Thank you for pointing out this discussion point. We have now addressed the result raised in the J Biol Chem 286: 12524–12532 in our discussion section (pg. 19). In our response we first acknowledge that previous work has shown HIF-1a activation could lead to a loss of GSIS. We subsequently argue that this difference may be due to the mode and intensity of HIF-1a activation. Where Sato et al. worked with varying oxygen tensions we worked with a level of hypoxia mimetic that was below a threshold for impact on beta-cell metabolism.

---

## [Decision Letter · Decision Letter 1]

29 Aug 2019

Hypoxia Induction in Cultured Pancreatic Islets Enhances Endothelial Cell Morphology and Survival while Maintaining Beta-cell Function

PONE-D-19-16565R1

Dear Dr. Rocheleau,

We are pleased to inform you that your manuscript has been judged scientifically suitable for publication and will be formally accepted for publication once it complies with all outstanding technical requirements.

With kind regards,

Feng Zhao

Academic Editor

PLOS ONE

---

## [Editor Report · Acceptance letter]

1 Oct 2019

PONE-D-19-16565R1 

Hypoxia Induction in Cultured Pancreatic Islets Enhances Endothelial Cell Morphology and Survival while Maintaining Beta-cell Function 

Dear Dr. Rocheleau:

I am pleased to inform you that your manuscript has been deemed suitable for publication in PLOS ONE. Congratulations! Your manuscript is now with our production department. 

With kind regards,

on behalf of

Dr. Feng Zhao 

Academic Editor

PLOS ONE